# The Antigenic Topology of Norovirus as Defined by B and T Cell Epitope Mapping: Implications for Universal Vaccines and Therapeutics

**DOI:** 10.3390/v11050432

**Published:** 2019-05-10

**Authors:** Jessica M. van Loben Sels, Kim Y. Green

**Affiliations:** Caliciviruses Section, Laboratory of Infectious Diseases, National Institute of Allergy and Infectious Diseases, National Institutes of Health, DHHS, Bethesda, MD 20892, USA; jessica.vanlobensels@nih.gov

**Keywords:** norovirus, adaptive immunity, monoclonal antibodies, epitope mapping, T cell epitopes, B cell epitopes, therapeutic antibodies, nanobodies (VHH), scFv, universal vaccines

## Abstract

Human norovirus (HuNoV) is the leading cause of acute nonbacterial gastroenteritis. Vaccine design has been confounded by the antigenic diversity of these viruses and a limited understanding of protective immunity. We reviewed 77 articles published since 1988 describing the isolation, function, and mapping of 307 unique monoclonal antibodies directed against B cell epitopes of human and murine noroviruses representing diverse Genogroups (G). Of these antibodies, 91, 153, 21, and 42 were reported as GI-specific, GII-specific, MNV GV-specific, and G cross-reactive, respectively. Our goal was to reconstruct the antigenic topology of noroviruses in relationship to mapped epitopes with potential for therapeutic use or inclusion in universal vaccines. Furthermore, we reviewed seven published studies of norovirus T cell epitopes that identified 18 unique peptide sequences with CD4- or CD8-stimulating activity. Both the protruding (P) and shell (S) domains of the major capsid protein VP1 contained B and T cell epitopes, with the majority of neutralizing and HBGA-blocking B cell epitopes mapping in or proximal to the surface-exposed P2 region of the P domain. The majority of broadly reactive B and T cell epitopes mapped to the S and P1 arm of the P domain. Taken together, this atlas of mapped B and T cell epitopes offers insight into the promises and challenges of designing universal vaccines and immunotherapy for the noroviruses.

## 1. Introduction

Human norovirus (HuNoV) is a major cause of acute gastroenteritis and has emerged as the leading cause of severe childhood diarrhea in populations vaccinated against rotavirus [1,2]. Norovirus is transmitted by the fecal‒oral route, usually through ingestion of contaminated food or water or by direct contact with an infected individual [3]. The disease is largely self-limiting in healthy individuals, with symptoms appearing as early as 15 h post-infection and lasting 24–72 h [4,5]. Virus can be shed in stool as late as 30 days following the initial infection, facilitating further spread [6]. With a low infectious dosage of ≤20 viral particles [7], norovirus is easily transmitted within confined areas such as homes, schools, hospitals, and cruise ships.

Norovirus disease is associated with a higher risk of life-threatening dehydration in the young and old, immunocompromised individuals, and those with limited access to health care [8,9,10,11]. The CDC estimates that the U.S. population experiences 19–21 million norovirus illnesses each year, with an estimated 570 deaths in children [12,13]. On a global scale, HuNoV has been estimated to cause approximately 200,000 deaths each year in children under five years old [1]. The financial toll associated with HuNoV infections is estimated to be $2 billion per year in the USA and $60 billion globally [12,14]. These statistics support the need for vaccines and therapeutics that would reduce norovirus morbidity and mortality in at-risk populations and lessen the social and economic impact.

Several factors have hindered the development of broadly protective HuNoV vaccines and treatment, including the marked antigenic diversity of norovirus strains and an incomplete understanding of protective immunity [15]. This review aims to summarize the collective data of norovirus epitope mapping studies in order to elucidate common features of strain-specific and cross-reactive antigenic sites. Current approaches to the characterization of norovirus monoclonal antibodies (mAbs) with regard to specificity, function, and mapping are presented for over 300 B and 18 T cell epitopes.

## 2. Antigenic Diversity of Noroviruses

Noroviruses are classified in the family *Caliciviridae.* They are small (30–40 nm), nonenveloped viruses with a single-stranded, positive sense RNA genome that is organized into three open reading frames (ORFs) (Figure 1A). ORF1 encodes a nonstructural polyprotein that is proteolytically processed during replication into six proteins, which includes the viral RNA-dependent RNA polymerase [16].

The major structural capsid protein, VP1, and the minor structural protein, VP2, are encoded by ORF2 and ORF3, respectively [19]. VP1 monomers assemble into 90 dimers that interact with VP2 to create the complete proteinaceous capsid [17]. Structural studies of VP1 have defined two major domains [17] (Figure 1A). The N-terminal shell (S) domain forms the innermost layer of the viral capsid and acts as a scaffold to surround the RNA genome [17]. The protruding (P) domain is linked to the S domain via a flexible hinge region [20], and the P domain is further divided into P1 and P2 subdomains [17]. The S domain retains the highest degree of genetic conservation, whereas the P2 subdomain is the least conserved. The P2 domain contains six flexible loops (Figure 1B) and has been implicated in differential ligand binding and antigenicity [18,21,22]. 

Noroviruses are classified into seven genogroups (GI though GVII), which are further subdivided into more than 40 genotypes [23,24,25] (Figure 1C). Norwalk virus (NV) is the prototype virus for the genus *Norovirus* and has been assigned genotype GI.1. VP1 sequences can vary up to 60% between genogroups and up to 30% between genotypes, but this genetic diversity has not yet been correlated with a classical serotyping system based on neutralization. In addition to genotypic diversity, noroviruses, like other RNA viruses, undergo error-prone replication, which results in the generation of highly diverse viral RNA populations [26,27]. This high mutation rate allows noroviruses to adapt rapidly to changing selective pressures and thus serves as an important mechanism to achieve immune evasion. 

Most human norovirus infections are caused by GI or GII genotypes, with fewer cases of GIV. The majority of HuNoV outbreaks and sporadic illnesses over the past few decades have been caused by a single genotype, GII.4 [28,29,30,31]. Several studies have described the selective pressure of herd immunity, which appears to drive an epochal-type evolution of GII.4 through antigenic drift [32,33,34,35,36,37,38]. The periodic emergence of new pandemic GII.4 antigenic variants has been linked, in part, to evolution in defined epitopes in VP1, likely involved in neutralization [35,38]. Though GII.4 continues to be the predominant genotype, an unexpectedly high number of global outbreaks caused by GII.17 in 2014 and 2015 raised the possibility that GII.4 could at some point be displaced [39,40,41,42]. Genetic diversity and varying patterns of evolution among the norovirus genotypes complicate vaccine development [43]. Sustained efforts to understand the relationship between norovirus diversity and immunity, and the mechanisms by which noroviruses evade this immunity are needed. This review summarizes progress in these areas that are attributable to nearly three decades of epitope mapping studies. 

## 3. Adaptive Immunity to Noroviruses

Difficulties in the efficiency of cell culture systems and animal models have historically limited an in-depth analysis of HuNoV immunity and viral escape [44]. Murine norovirus (MNV), which replicates efficiently both in vitro and in vivo, has been studied extensively to establish a proposed model of intestinal immunity in mice (Figure 2). There is evidence that MNV strains are endocytosed through M cells into the lamina propria of the small intestine, where they migrate to mesenteric lymph nodes (MLN) and gut-associated lymphoid tissue (GALT) [45,46,47]. Peyer’s patches, which sample antigens from the lumen for immune surveillance, contain high titers of MNV [46]. Once MNV breaches the epithelial barrier, neutrophil and mononuclear cell infiltration increase within Peyer’s patches, and MNV proceeds to infect immune cells of myeloid and lymphoid origin, including macrophages, dendritic cells, B cells, and T cells [46,48,49,50]. 

Interferons (IFNs) are released by infected cells during the initial innate immune response and play a key role in limiting MNV replication [51,52,53]. However, the IFN response alone is often insufficient to protect against acute norovirus infection in mice. Full MNV clearance has been achieved by the administration of activated T and B cells in mice [54], illustrating the importance of adaptive immunity in controlling infection. Several studies with MNV have observed increased levels of cytokines and chemokines related to activation and trafficking of CD4+ helper T (Th) cells, with an emphasis on Th1 responses [55,56]. Increases in IFN-γ, IL-2, TNF-α, MIP-1, and granzyme-B have been associated with the activation of cytotoxic T lymphocytes (CTLs), which were responsible for the lysis of infected cells [55,57,58]. Activated CD8+ T cells and increased IFN-λ in particular have significantly reduced viral loads and helped clear persistent MNV infection in mice [58,59,60]. Inefficient acquisition and maintenance of polyfunctional T cell capability with respect to cytokines, chemokines, and cytotoxic potential have been associated with persistent MNV infection [58]. CD4+ T cells were correlated with protection upon re-exposure to MNV in mice [61].

Humoral immunity supplements the primary Th1 response and plays a critical role in MNV clearance and protection from subsequent infection. Mice incapable of expressing MHC II could not mount a protective immune response, indicating that Th2 responses, which are responsible for full B cell activation and maturation, were also necessary for controlling MNV infection [61]. MNV-specific antibodies were also shown to reduce the systemic spread of MNV [45]. B cells and antibody-secreting cells (ASCs), known to migrate to the gut and confer protection against an array of intestinal pathogens [64], were found in the intestinal tissues of mice infected with MNV [46]. Additionally, passive treatment of mice with neutralizing monoclonal antibodies was shown to control MNV infection in systemic sites as well as in the intestine and MLN [49,65].

Although it has been reported that MNV and HuNoV share common features in structure and replication [50,66], it has been challenging to extrapolate a number of findings in the mouse model directly to humans. There are noteworthy differences between the two virus groups. Murine noroviruses are less diverse genetically, constituting a single genotype [67]. Human noroviruses replicate in intestinal enterocytes, both in vitro and in vivo, a cellular tropism that MNV does not share [68,69]. Another important difference resides in the carbohydrates used as attachment factors: MNV utilizes sialic acid while most HuNoV recognize HBGA carbohydrates [70,71]. The cellular receptor for MNV, CD300lf, is expressed on immune cells and Tuft epithelial cells, known permissive cell types for MNV infection [62,72]. A proteinaceous receptor for HuNoV has not yet been identified. While these differences exist and must be considered, observations in wild type and genetically modified mice have provided context in the investigation of protective immune mechanisms for noroviruses.

During human norovirus infection, monocyte recruitment and cytokine activation have been detected [5,73,74,75,76]. In particular, an increase in serum IFN-γ, IL-2, and TNF-α levels was found soon after infection, consistent with an upregulation in cytotoxic CD8+ T cell immunity facilitated by Th1 cells [5,73,77,78]. Regulation of this cytotoxic response via IL-10 and chemokine release, as well as a prolonged regulatory T cell response, has been reported in several studies [5,73]. Additionally, Th2 cytokines IL-4, IL-5, IL-6, and IL-8 were detected as early as two days post-infection [73,76]. This CD4+ Th2 response plays a critical role in B cell activation and the establishment of humoral memory [61,76,79], though overall cellular immunity appears weighted toward a Th1 response [5,76]. Cellular immunity requires further investigation in humans, as most of these findings have been determined using peripheral blood T cells, which may differ from the immune cells found at the site of infection.

Humoral immunity to HuNoV has been studied in far more depth, and is considered stronger and more long-lasting than T cell immunity [78,80,81,82]. An estimated 90% of the adult population is seropositive to norovirus [83]. Though likely comprising only a small fraction of the total HuNoV-specific antibody population in serum, protective antibodies mediate reduced infection and severity of gastroenteritis [84,85,86,87,88]. Consistent with the importance of antibodies, B cell dysfunction has been associated with the establishment of chronic HuNoV infection [9,89]. Chronic norovirus infection in one such patient was resolved following the gradual reconstitution of functional B cells after chemotherapy and development of protective HBGA blocking antibodies [90]. In this report, we review approximately 300 monoclonal antibodies (mAbs) representing distinct epitopes that span the VP1 capsid protein of human GI and GII genotypes as well as murine GV strains. Of these mAbs, 123 have demonstrated evidence of protective capacity in surrogate, in vitro, or in vivo neutralization assays. These mAbs have defined immunologically important epitopes and provided insight to adaptive immune responses to norovirus infections. These findings will be summarized below.

## 4. Isolation and Characterization of Norovirus Monoclonal Antibodies

### 4.1. Recombinant Norovirus Capsid Antigens

Virus-like particles (VLPs) have largely informed immunological studies of HuNoV. VLPs are replication-deficient, empty capsids that are self-assembling and antigenically indistinguishable from live virus [91,92]. Norovirus capsid proteins have been produced in bacteria, plants, insect cells, and mammalian cells [93,94,95,96,97]. Baculovirus- and Venezuelan Equine Encephalitis virus (VEE) replicon-based methods are the most widely-used production systems for HuNoV VLPs. Both methods produced VLPs of the same size and shape, but subtle differences were reported in epitope availability and temperature sensitivity of the VLPs [98]. Accessibility of certain mAbs to their cognate epitopes was affected by factors beyond primary amino acid sequence, including the temperature and pH at which the VLPs were assembled, and post-translational modifications [98,99]. In addition to origin-based variability in VLPs, certain assay conditions can affect epitope presentation. Plastic-bound VLPs in direct ELISAs were found to display more cryptic epitopes than VLPs analyzed in indirect ELISAs or surface plasmon resonance (SPR) [100]. Temperature shifts during binding assays were also found to affect differential ligand binding [33,78,101]. The type and quality of carbohydrates used in surrogate neutralization assays may also affect results [102]. Therefore, the characterization of HuNoV epitopes should consider these technical challenges and explore binding under varying conditions.

Modified VLPs have been used extensively to map epitopes via site-directed mutagenesis, P2 domain swaps, and chimeric VP1 proteins [36,88,103,104,105,106]. Panels of time-ordered VLPs proved to be especially useful in monitoring the evolution of GII.4 variants in context of herd immunity [36,98,107]. In addition to full VLPs, HuNoV P particles have become an important tool in epitope characterization. These subunit particles are produced by expression of the P domain of VP1 in easily scalable bacteria or yeast systems, whereby monomers can dimerize or self-assemble into 12mer or 24mer particles [108,109,110,111]. P particles bind the same ligands as full VLPs, though some critical blockade epitopes appear to have different conformations [36]. Other comparative studies of P particles and VLPs have reported variable results in the ability of these two antigens to stimulate B and T cell responses [112,113].

### 4.2. Origins and Types of Norovirus Monoclonal Antibodies

Monoclonal antibodies (mAbs) have become essential tools for defining immunologically relevant norovirus epitopes. Appendix A summarize over 70 published studies that have reported the isolation and characterization of mAbs directed against HuNoV or MNV. The immunogen(s) and human or animal source used in the development of each mAb are indicated. The strain specificity, conformational or linear nature, activity in relevant functional assays, and mapped amino acid residues associated with each epitope are included. It should be noted that throughout this review, an “epitope” is considered defined by the continuous or non-continuous residues of VP1 that interact directly with the mAb. Mapping data that includes structural analyses of the antibody and viral antigen complex are required for optimal precision. However, mapping studies may report residues that indirectly affect the presentation and recognition of an epitope, and these published data are included in the tables. A total of 307 unique mAbs are represented in the four tables, consisting of 91, 153, 21, and 42 GI-specific, GII-specific, MNV GV-specific, and cross-genogroup reactive mAbs, respectively.

The majority of norovirus mAbs (228 mAbs) have been isolated from naïve animals initially hyperimmunized with a single VLP or live strain and followed with various boosting and adjuvant regimens. The antigens and strategies used for screening or panning antibodies play important roles in the selection process. Immunizations and screens with GI VLPs, primarily NV, yielded mAbs that predominantly bound homotypic VLPs (Appendix A). Limited mAb cross-reactivity with other GI genotypes was observed (19 mAbs), and even fewer GI.1 mAbs recognized GII VLPs (11 mAbs). Exceptions included mAbs such as NV3901, NV3912, NS14, and NV23, which bind linear epitopes conserved across genogroups (Appendix A). Immunizations with GII VLPs, however, often gave rise to mAbs that recognized more than one GII genotype (Appendix A). Cross-reactive mAbs that recognize both GI and GII VLPs have been raised in animals immunized with a single VLP or with a mixture of heterologous VLPs (Appendix A). Immunizations with GIII bovine and GV murine norovirus antigens have also yielded mAbs that recognize human GI and GII genotypes [65,114,115,116]. Though the protective potential of these mAbs is largely unexplored, this panel of highly cross-reactive mAbs derived from diverse VLPs provides evidence of conserved B cell epitopes across norovirus genotypes.

Natural HuNoV infections stimulate polyclonal IgA, IgG, and IgM responses [117]. Though HuNoV is an enteric infection, most epitope mapping has been conducted with mAbs of the IgG class (176 IgG mAbs). Epitopes defined by IgG have proven to be biologically relevant because IV administration of neutralizing IgG, as noted above, limited MNV spread in mice [54]. Human IgG mAbs have also been generated from the PBMCs of naturally-infected patients, three of which showed GII.4 neutralization in vitro [118,119]. Additionally, the protective activity of HuNoV systemic antibodies may correlate with that of mucosal antibodies, as both antibody populations have shown ligand blockade activity in surrogate neutralization assays [120].

The tissue distribution of an antibody in vivo is arguably the most important attribute in determining protective potency [121]. Therefore, secretory IgA, which can be found in the gastrointestinal tract mucosa, is of special interest (Figure 3A). Far fewer monoclonal IgA antibodies have been isolated and characterized (20 mAbs total). IgA titers in both serum and saliva have been correlated with protection from GI.1 infection [121,122,123]. Human challenge with NV also revealed that a rapid, local mucosal IgA response was one of the few significant correlates of immunity [78,122]. Though often lower in titer, affinity, and avidity, IgA mAbs have demonstrated comparable potency to IgG in blocking the binding of HuNoV VLPs to ligands [124]. Based on studies with the human monoclonal IgA, mAb 5I2, the authors proposed that dimeric IgA blockade activity may have stemmed from its large molecular weight and steric hindrance of VLP binding to carbohydrates [125]. Additional monoclonal IgAs have been isolated from humans following natural infection and have neutralized live virus in vitro [118]. 

Newer antibody technologies have also proven useful in defining norovirus epitopes. Nanobodies (VHHs) are ~15 kDa, single-polypeptide chain antibodies derived from the unique variable heavy chain domains of camelid antibodies (Figure 3A). Immunization of camelids, namely llamas and alpacas, with HuNoV VLPs has led to the development of 29 HuNoV-specific nanoantibodies, or nanobodies [126,127,128,129,130]. Nanobodies have full binding capacity and affinities comparable to conventional antibodies [126]. Recombinant single chain variable fragments (scFv) are ~27 kDa recombinant proteins that often consist of the light and heavy chain variable regions of a monoclonal antibody separated by a flexible peptide linker (Figure 3A). Bypassing animal immunizations altogether, phage display technology allows the panning of an scFv library against an antigen of interest [131]. A total of 23 recombinant scFvs have been produced for HuNoV (Appendix A). Due to their smaller size, HuNoV epitope mapping with scFvs and VHHs has led to the characterization of several conserved regions buried within VLPs, which conventional mAbs have typically failed to recognize [129,130,132]. These mAbs have elucidated novel ways in which humoral immunity might be harnessed to neutralize infection and have therapeutic potential due to their versatility in genetic engineering and delivery to the site of infection. 

### 4.3. Binding, Blockade, and Neutralization Assays

Norovirus mAbs are subjected to a number of standard assays in their initial characterization such as ELISA and western blot analysis on denatured antigen. These assays help identify reactive regions and their respective conformations. Of the 307 mAbs reviewed here, 77 recognize linear epitopes that map to the S and P1 domains of both GI and GII genotypes (Appendix A). Though less conserved than S domain sequences, residues that make up the P1 domain are more readily accessible due to capsid flexibility [98]. Cross-reactive mAbs that map to linear epitopes have proven useful in diagnostic assays, but their protective capacity appears limited [132,133,134].

The majority of monoclonal antibodies have been selected to define HuNoV epitopes that are likely immunodominant, conformational, and neutralizing. In the absence of a cell-based neutralization assay for many years, surrogate neutralization assays were developed. While a proteinaceous receptor for HuNoV has not been identified, as it has for MNV [72], HuNoV VLPs were shown to bind to a wide array of histo-blood group antigen (HBGA) carbohydrates that function as attachment factors to cells in the gut [68,106,135]. The HBGA family includes Lewis antigens, secretor antigens, and ABO carbohydrates [136]. Four HBGA binding sites have been defined on a single VP1 dimer [137]. The susceptibility allele, *fut2*, is responsible for HBGA expression on cell surfaces and the secretion of soluble HBGAs in humans [68,122,138]. Individuals who express *fut2* are designated as “secretor-positive” and tend to be more susceptible to HuNoV infection [18,33,40,135,139,140,141]. However, some GI and GII genotypes can readily infect “secretor-negative” individuals and can bind to alternative carbohydrates [5,76,142,143,144]. HBGA binding residues vary greatly among HuNoV genotypes, and HBGAs themselves can take many alternative conformations and rotations [18,22,141]. Taken together, the relationship between HuNoVs and carbohydrate binding is complex and highly variable between strains.

One of the first HuNoV surrogate neutralization assays measured hemagglutination inhibition (HAI) [71] (Figure 3B). HAI antibody titers ≥40 were correlated with protection from NV-caused gastroenteritis in adult volunteer challenge studies [145]. While HAI continues to be utilized, the majority of mAb epitopes have been characterized using HBGA blocking or “blockade” assays (Figure 3C). Antibodies with blockade activity inhibit the binding of VLPs to natural HBGA molecules in saliva, synthetic HBGAs, or HBGA-containing pig gastric mucin [146]. Blockade titers ≥200 and titer increases ≥4-fold have been associated with increased protection against HuNoV challenge [36,139,147,148]. Some mAbs tested for both HAI and HBGA blockade have demonstrated different activities in the two assays [36,102,149]. These observations suggest there may be multiple ligand interactions involved in HuNoV infection [102], but ultimately, variability in these quantitative assays may simply reflect their surrogate nature for the measurement of true neutralization.

Antibody neutralization is traditionally defined with in vitro or in vivo studies and inhibition of live virus infections. Neutralization activity of NV mAbs in the chimpanzee model was shown by pre-incubation of infectious virus with antibodies prior to challenge, but this animal model is no longer available [87,88]. For in vitro studies, an organoid enteric epithelial cell culture system has recently shown success [68,119]. Though preferential towards specific strains, this system was the first to demonstrate neutralization of live HuNoV in cell culture [118,119]. HBGA expression in intestinal enteroid cells (i.e., a secretor-positive phenotype) significantly enhanced viral attachment and replication for certain noroviruses [68]. Therefore, antibodies that block HBGA interactions may well play an instrumental role in humoral protection. The in vitro neutralization assay in enteroids proved to be more sensitive than HBGA blockade and HAI for certain mAbs [118]. As this in vitro system is optimized and expanded to support the replication of more HuNoV genotypes, it will be possible to evaluate additional antibodies for neutralizing activity.

### 4.4. Precision B Cell Epitope Mapping

Mapping of conformational mAb binding sites has been conducted using various mutagenized or chimeric HuNoV VLPs in HBGA blockade assays. By monitoring VP1 residue changes that result in the loss of blockade activity, a minimal binding epitope for an mAb can be assigned. GII.4 epitopes have been mapped with VLPs that represent naturally-circulating pandemic strains, as well as those from immunocompromised patients with chronic norovirus infection [9,10]. Chronic norovirus infection can extend for years in the absence of immune clearance, with the accumulation of diverse viral RNA populations [9,150,151,152]. Analysis of these in vivo-evolved strains, possibly selected in the presence of therapeutic pooled immunoglobulin treatment, have allowed for the discovery and characterization of HBGA blockade epitopes that were computationally predicted, yet unconfirmed [153,154].

Structural studies have modeled the precise interactions between mAbs and their epitopes in VP1 [18,135,141,155,156,157]. Several mAbs have been co-crystallized with HuNoV P domains [41,119,125,130]. The fine mapping data acquired in structural studies were generally consistent with earlier epitope mapping studies that had incorporated analysis of antibody function, mutagenesis, and sequence alignments. These studies had led to the prediction of several HBGA blockade epitopes on GII.4 strains [36,158,159], four of which were subsequently confirmed in empirical studies [33,103,146]. Continued structural studies of these mAbs and others will be important in refining the antigenic topology of noroviruses with increased precision.

## 5. HBGA Blockade and Human Norovirus Monoclonal Antibodies

### 5.1. Mechanisms of Antibody Blockade

Blockade mAbs have been correlated with reduction in the severity of gastroenteritis in challenged individuals and therefore likely play a critical role in HuNoV immunity [139]. The blockade potency of an mAb is determined by two factors: i) the amino acid residues that comprise the mAb epitope and ii) the flexibility of the structure of the P domain that controls epitope access. 

The majority of HBGA blockade epitopes are conformational and include residues that are within or proximal to HBGA binding sites. Intuitively, direct binding of a mAb to the HBGA binding pocket would prevent viral interaction with HBGAs. Monoclonal antibodies that bind residues immediately adjacent to an HBGA binding site could prevent ligand interaction via steric hindrance by the bulky antibody itself. Antigenic drift in the P domain can overcome both blockade mechanisms by change in the mAb epitope sequence. Over years of selective pressure, amino acids surrounding an HBGA binding domain tend to undergo more antigenic drift than the binding pockets themselves [38]. This results in the loss of mAb binding without change to the critical HBGA binding residues (Figure 3C). In addition, amino acid changes adjacent to blockade epitopes can prevent mAb binding through conformational occlusion [160]. By changing the charge or size of adjacent amino acids, mAbs can be sterically prevented from binding to their cognate epitopes (Figure 3C). The probability of antigenic drift resulting in an occluded epitope is likely determined by its location and function on the virion [160]. The evolution of residues in and around the HBGA binding pocket can also influence HuNoV binding patterns [33,36]. It has been proposed that such mechanisms could expand the pool of susceptible hosts for a particular norovirus strain [125,135].

The flexibility of the norovirus capsid likely complements antigenic drift in the evasion of humoral immunity. Conserved epitopes found buried within the VLP structure are often occluded at room temperature and become exposed at 37 °C [98] (Figure 3C). Occlusion of critical epitopes at less favorable temperatures may protect the virus from the external environment and from degradation during transmission [98]. Global particle structure, termed “viral breathing,” is regulated by amino acids that comprise a proposed viral “breathing core” [160]. This may include a set of residues called the NERK motif (amino acids 310, 316, 484, and 493), with residue 310 implicated in the emergence of new pandemic GII.4 strains [98]. The P domain loops in the particle also demonstrate significant flexibility that can regulate epitope presentation and ligand binding [160]. Taken together, the dynamic states of VP1 and fully-assembled VLPs play instrumental roles in HuNoV antigenicity, ligand binding, and likely infection.

Particle flexibility can be both useful and detrimental to mAb HBGA blockade. mAbs can confer blockade indirectly by binding residues distant from the HBGA binding pocket that result in allosteric changes to the P domain [160,161]. However, antigenic drift and conformational occlusion can result in the loss of mAb binding, or force the P domain into a conformation that lessens allosteric HBGA blockade [161,162]. Exposure of virions to body temperature during infection may affect the binding to blockade antibodies [160]. Moreover, “viral breathing” can camouflage certain epitopes through changes in the “breathing core” (Figure 3C). One interesting theory postulates that HuNoV might interact with soluble HBGAs prior to cell attachment, resulting in conformational changes in the P domain that would occlude neutralizing antibody access while maintaining the ability to attach [119]. These camouflaged epitopes may be interesting targets in future vaccine and therapy designs.

### 5.2. Blockade Epitopes of Genogroup I Noroviruses

Of the 91 mAbs that demonstrate GI-specific reactivity, 62 have been tested for HBGA blockade (Appendix A). Most GI blockade mAbs isolated thus far are GI.1-specific, with a few GI.3 and GI.4 mAbs described [121]. GI.1 NV blockade epitopes in the P domain are spatially distinct from non-blockade epitopes, and two mAbs (D8 and B7) have demonstrated protection from NV challenge in chimpanzees [88,121]. Two GI blockade epitopes have been mapped with mAbs 54.6 and 5I2 (Figure 4A). mAb 54.6, an IgG antibody raised in mice immunized with GI.1 NV, demonstrated HAI activity and reduced VLP binding to Caco-2 cells [163]. This conformational epitope was predicted to include residues 280 and 291–293, and possibly 302 [164]. The structural analysis of IgA MAb 5I2 binding represented the first mapped IgA epitope for any HuNoV. Generated from human PBMCs following GI.1 challenge, mAb 5I2 blocked NV HBGA binding and inhibited hemagglutination [124]. Structural analysis revealed that its conformational binding epitope consisted of residues in loops T, U, and Q [125]. Both mAbs appeared to inhibit HBGA binding through steric hindrance [125,164,165]. While many GI mAbs have been tested for blockade or HAI, most (57 GI mAbs) have not been defined at the structural level.

Genogroup I nanobodies have recently been generated and mapped to their corresponding epitopes. Nano-7 and Nano-64 bound the GI.1 dimer interface, while Nano-94 bound an epitope shared with mAb 54.6 (Figure 4A). Nano-7 and Nano-94 also demonstrated significant HBGA blockade activity. Nano-94 caused VLP aggregation, while Nano-7 showed HBGA blockade activity through minor structural rearrangements in the P domain [130]. Like most other GI-specific mAbs, these nanobodies failed to bind other GI genotypes. This genotypic restriction is likely due to sequence diversity across the genogroup, resulting in structural differences in the P2 domain [78,130]. 

### 5.3. Blockade Epitopes of Genogroup II Noroviruses

Several blockade epitopes have been successfully characterized for genotypes GII.2, GII.4, and GII.10 (Appendix A). A majority, however, have been mapped to the GII.4 genotype. There are 8 predicted or mAb-mapped GII.4 blockade epitopes, labeled Epitopes A through H (Figure 4B). Epitopes A through E were predicted based on sequence alignments and comparison of variable residues across pandemic GII.4 sequences [36]. Putative Epitope B (residues 333 and 382) is buried in the dimer interface between two VP1 monomers and is thought to influence exposure of residues located on the surface of VLPs. Epitope C (residues 340 and 376) is located on the surface and lateral edge of the capsid, directly proximal to the HBGA binding pocket. No blockade mAbs have been mapped to Epitope B. In contrast, VHH mAb M7 has been mapped to the Epitope C region of GII.4-MD2004 and showed strain-specific HBGA blockade activity [102]. This finding validated epitope prediction efforts and the strain specificity of Epitope C, though further epitope characterization is needed.

The hypervariable Epitope A (residues 294–298, 368, 372, 373) has demonstrated significant immunodominance compared to other GII.4 blockade epitopes. Although a GII.4 virion is predicted to have the same number of Epitopes A, E, F, and G, ~40% of serum blockade responses target Epitope A [37,103]. Similarly, administration of a multivalent VLP vaccine in humans resulted in a memory immune response that predominantly recalled Epitope A from a previously circulating GII.4 strain [86]. Evolution within Epitope A has correlated directly with new strain emergence. Polymorphisms at residues 294, 297, 368, and 372 in particular appear to have driven the evolution of New Orleans 2009 and Sydney 2012 strains [37,158]. Epitope A faces the exterior of the VLP in the P2 domain and spans loops A and B [34,104,166]. The presentation of Epitope A does not appear dependent on particle conformation, as changes in the NERK motif did not affect blockade potency of mAbs mapped to this epitope [98]. Therefore, antigenic drift appears to be the driving mechanism of viral immune evasion at Epitope A. 

Epitope D (residues 391, 393–396) mediates the binding of both mAbs and HBGAs directly. Like Epitope A, Epitope D faces the exterior of the VLP in the P2 domain and is unaffected by particle conformation caused by changes to the NERK domain [98]. Residues in Epitope D map to loops T and U and are located along the ridge of the HBGA binding domain. This positioning makes these residues fully available for mAb binding [36,166]. Polymorphisms at Epitope D, especially 393, have been implicated in escape from herd immunity as well as HBGA ligand switching [36,38,167]. Although not part of the HBGA binding pocket itself, changes in Epitope D can modulate affinity for different HBGAs by stabilizing bonds with non-H antigen HBGAs [125,135]. Residue variation within Epitope D of pandemic GII.4 strains New Orleans 2009 and Sydney 2012 and emerging GII.17 strains have correlated with the loss of blockade potency of several mAbs [38,167]. Amino acid variations seem to be well-tolerated, as HBGA binding can be maintained with various residues while also driving escape from neutralizing mAb binding [154]. GII.4 mAbs that map to Epitope A and D (29 and 5 mAbs, respectively) have consistently been identified in studies from several independent research groups (Appendix A).

Epitope E (residues 407, 412, and 413) maps to loops T and U in the P domain and is lateral to Epitopes A and D on the outermost surface of the VLP. However, Epitope E is less surface exposed than either Epitope A or D [36]. Residues in Epitope E have varied with every major GII.4 epidemic after 2002, suggesting that it is a hot spot for the emergence of immunologically novel GII.4 strains [36,146]. Only a single mAb has mapped to Epitope E, possibly illustrating how particle conformation can influence the antigenicity of this epitope. mAb GII.4E was raised in a mouse immunized with GII.4-2002 VLPs and demonstrated strain-specific blockade [146,166]. Lower temperatures and amino acid changes around the epitope were predicted to occlude GII.4E binding [160]. Located near the P2-P1 C-terminal boundary, this temperature-dependent epitope is evidence of immunologically significant epitopes hidden in the interior of the VLP structure.

Similarly, access to Epitope F (residues 327 and 404) is temperature-dependent and can be influenced by mutations in the NERK domain [160]. Epitope F is conserved across all GII.4 strains, and human mAb GII.4F has been mapped to this HBGA blockade epitope [36,98]. However, changes at residue 310 have altered Epitope F protection through an unknown, allosteric mechanism. Additionally, residue 234 works in concert with NERK to regulate global particle structure and consequently mAb conformational occlusion at this epitope [36,86]. Interestingly, changes in Epitope F have affected the mAb binding to proximal Epitope E [160]. Though these epitopes do not share the same residues, it appears that Epitope F regulates mAb access to Epitope E via changes in local particle conformation [98]. 

The remaining lettered GII.4 epitopes have been mapped with a single mAb, yet their exact epitope residues are uncharacterized. Epitope G is hypothesized to be nearby, if not overlapping with, Epitope F [98]. mAb GII.4G, raised in a mouse immunized with GII.4-2002, has exhibited HBGA blockade against GII.4-2009 and GII.4-2012 [37]. GII.4G blockade also appears to be conformation- and temperature-dependent [98,160]. Finally, the undefined Epitope H was detected by mAb GII.4-2012-G8. This mAb was generated from a mouse immunized with the latest GII.4 pandemic strain, Sydney 2012. GII.4-2012-G8 binding to Sydney 2012 VLPs has demonstrated HBGA blockade activity, but at an unmapped epitope that appears to be influenced by Epitope A and the residue R373 [168].

Several mAbs block GII.4 HBGA binding at epitopes not included in this lettered system. VHH M6, for example, exhibited cross-reactivity to 7 different GII genotypes as well as blockade against GII.4 MD145 VLPs. Unlike other HBGA blockade mAbs, M6 mapped to a linear epitope located on the C-terminal P1 domain [102]. The mechanism by which M6 blocks HBGA binding may be strain-specific or unique to VHHs, as other mAbs also bind this region yet fail to block HBGA binding [127]. Additionally, 2C3G3 blocked binding of 2006b VLPs at a conformational epitope that appears to also modulate HBGA interactions [169] (Figure 4B). Four GII.4 blockade mAbs have mapped to conformational, quaternary structures, though these precise residues have not been determined [118]. Several VHHs have blockade activity against GII.4 and GII.10 epitopes and exhibited similar activity when the P particle was dimeric, but not monomeric [129] (Figure 4C). Taken together, mAbs have played vital roles in understanding mechanisms of HuNoV immune evasion and in the identification of immunodominant epitopes. However, further characterization of conserved epitopes, especially those corresponding to highly cross-reactive mAbs against heterotypic genotypes, is needed. Translating these epitopes to less-studied genotypes (i.e., those outside of GI.1, GII.4, and GII.10) would begin to address the possibility of targeting cross-protective B cell epitopes.

## 6. Beyond Blockade: Virus Neutralization

### 6.1. Neutralization of Murine and Human Noroviruses

Murine norovirus replicates efficiently in cell culture and classical viral neutralization assays such as plaque reduction can be applied to the analysis of antibodies. The MNV receptor CD300lf [72] is a cell surface molecule that binds a wide range of lipids on its extracellular immunoglobulin domain [170]. Structural studies have shown that the A’B’ and D’E’ loops in the MNV P2 domain engage with CD300lf, initiating cell attachment and viral infection [171,172]. Recently, metal ions and bile acids have been shown to enhance receptor binding and increase infectivity of MNV by regulating conformation of the P domain as it interacts with CD300lf [172]. Bile acids enhance replication of certain HuNoV genotypes in intestinal enteroids by an unknown mechanism and in other genotypes, appears to stabilize HBGA interactions [68,173]. The entry of HuNoV has not yet been linked to a proteinaceous receptor, complicating direct comparisons in the mechanisms of neutralization. 

A repertoire of mAbs has been generated against MNV (21 mAbs, Appendix A). The earliest characterized MNV-specific IgG, MAb A6.2, neutralizes viral entry by binding to an epitope that spans the A’B’ and C’D’ loops of the P domain [162,174] (Figure 5A). This epitope overlaps with the CD300lf binding site, thereby blocking viral attachment directly [172]. Additionally, MAb A6.2 binding limits the flexibility of the P2 domain and drives the receptor domain into a weak-binding conformation [162]. However, this mAb exhibited limited cross-neutralization, and neutralization escape was quickly achieved within five passages of MNV in the presence of the antibody in vitro [65,175]. In contrast, the monoclonal IgA 2D3 neutralized 10 strains of MNV, including two persistent mAb binding sites [65,161,162]. This indicates that the flexibility of the P domain across most norovirus genogroups likely plays a critical role in cell attachment and immune evasion. 

With the very recent establishment of the intestinal enteroid system, the majority of HuNoV blockade mAbs have not yet been validated for neutralization in vitro. However, a study in 2018 generated and characterized 25 IgG and IgA mAbs from naturally-primed human PBMCs [118]. Twenty mAbs mapped to three major antigenic sites on the P domain, and 14 mAbs exhibited HAI and/or HBGA blockade. Five blockade mAbs were selected from different antigenic groups and successfully neutralized live GII.4 infections in vitro. The epitopes of these neutralizing mAbs were not mapped, but this was the first demonstration of the link between HBGA blockade and neutralization in humans. Only a single HuNoV mAb to date with both HBGA blockade and neutralization activity has been mapped [119] (Figure 5B). The 10E9 mAb prevented GII.4 Saga-2006 VLPs from binding HBGAs and neutralized patient-derived GII.4 viruses in vitro. Co-crystallization structural studies showed that the epitope overlapped the HBGA binding pocket, reminiscent of MNV-neutralizing mAbs that bound to the CD300lf pocket. Taken together, these studies provide evidence that mAbs with HBGA blockade and HAI activity can have the capacity to neutralize live HuNoV in vitro. 

### 6.2. Beyond Receptor Binding Inhibition: Other Mechanisms of Neutralization

Antigenic diversity is the hallmark of the noroviruses. However, the maintenance of structural integrity and key capsid functions such as receptor binding undoubtedly require certain sequences to remain constant over time. For human noroviruses, the ability to target conserved epitopes buried within the VLP structure offers insight into mechanisms of antibody-mediated neutralization involving such conserved sequences. Nanobodies in particular have begun to define these mechanisms. Depending on the VHH treatment, nanobody binding has resulted in the aggregation, disassembly, and structural rearrangement of VLPs from several HuNoV genotypes [129,130]. Thus, neutralization can be achieved by compromising overall capsid morphology and integrity. 

Several mechanisms of particle reorganization have been characterized using VHHs. Treatment with GII.10-specific Nano-32 produced large aggregates of apparently intact VLPs when viewed under EM. Nano-32 binding induced a conformational rearrangement of several P domain loops, thereby altering the hydrophobic landscape of the P domain surface [129]. Nano-94 also caused GI.1 particle aggregation, but through an uncharacterized mechanism [130]. Nano-26-driven disassembly was thought to involve the engagement of a conformational epitope that spanned the GII P dimer interface. Nano-26 binding resulted in the stabilization of an unfavorable dimer conformation, thereby preventing HBGA binding [129]. GI-specific Nano-62 recognized an epitope that was similar to Nano-26 yet failed to induce particle disassembly. Nanobody-specific interactions and orientation with respect to the P domain may explain this difference in the effect on viral integrity [130]. Nano-85, in contrast, bound a temperature-sensitive, conserved epitope located in the GII C-terminal P1 domain. Upon epitope engagement, Nano-85 was thought to act as a fulcrum between the S and P domains to disrupt overall particle structure. This mechanism was proposed to mimic natural viral disassembly and release of viral RNA upon infection [127,129]. Several GII.4 IgG mAbs have also been suggested to target viral entry and uncoating mechanisms, though this was largely speculative [98]. 

Appendix A records published cross-reactive mAbs, most of which have not been tested for HBGA blockade. This includes some scFvs that were isolated via carbohydrate elution, thereby enriching antibody-phages that bind at or near HBGA binding sites [176]. However, studies have reported that <1% of antibodies in convalescent sera have HBGA blocking activity, which suggests that these non-blockade antibodies may also be important in protection [177,178,179]. Though several cross-reactive mAbs bind non-blockade epitopes, they could potentially neutralize through mechanisms such as antibody-dependent complement lysis and opsonization. In vitro and in vivo neutralization assays can help characterize the immunological relevance of non-blockade mAbs. Because HuNoV infection appears to elicit a degree of cross-genotypic protection [76,180,181,182], further investigation will determine if the cross-reactive mAbs summarized here can help elucidate the mechanisms of heterotypic immunity.

## 7. T Cell Epitope Mapping

Antigenic variation of HuNoV has clearly challenged humoral immunity in the establishment of long-lasting cross-protection. Similarly, HuNoV diversity may also confound T cell-mediated immunity. Human challenge studies found evidence of cross-reactive T cell responses to heterologous VLPs in the form of an increase in IFN-γ [76,78]. However, different strains of MNV induced various levels of protective immunity, which was attributed to interference with T cell activation and IFN release by APCs, difference in capsid sequences, or immunity antagonized by the MNV-specific protein, VF-1 [61,183]. A cytokine study in humans similarly proposed that varying T cell activation had occurred based on the infecting strain [73]. In the gnotobiotic pig HuNoV challenge model, a P particle vaccine candidate induced strong CD4+ responses in all tissues as well as IFN-γ-secreting CD8+ T cells in the intestine [184]. The identification of epitopes that elicit strong T cell responses to diverse strains will be an important advance. 

T cells interact with MHC proteins on infected cells or professional antigen-presenting cells (APCs), which present short peptides from digested intracellular proteins. While a single antigen-presenting cell may present many peptides simultaneously on various MHCs, T cell recognition is restricted to a single peptide. The antigenic diversity of norovirus and the relative infrequency of conserved amino acid sequences may directly limit the ability of T cells to establish a cross-reactive memory response. Additionally, it has been suggested that MHCs may preferentially present peptides that are not conserved between different genotypes [76]. This, in turn, limits protection against reinfection and reduces cross-reactive cytotoxic T lymphocyte activity and antibody-mediated immunity. 

There have been only a few T cell epitopes identified for norovirus (Appendix A, Figure 6). Most T cell epitope mapping has been carried out with peripheral T cells isolated from VLP immunized- or MNV infected-mice. Splenocytes were harvested, sorted, and stimulated with short, overlapping peptides. Increased cytokine release marked T cell activation. Immunization with MNV led to the characterization of several epitopes that, based on sequence homology, are predicted to share cross-reactivity with HuNoV. Of particular note were T cell epitopes that mapped to VP1 amino acids 461–473 and 519–527, regions of the P1 domain that are conserved across GII and GI-GV genogroups, respectively [58,185]. However, cross-reactivity against HuNoV genotypes has only been confirmed experimentally for a few T cell epitopes. One study described two GII-specific, two GII.4-specific, and two GII.4-1999 strain-specific epitopes spanning VP1 that induced T cell immunity in mice [177]. Both CD4+ and CD8+ T cells recognized these VP1 peptides in the context of MHC molecules and upregulated IFN-γ production, often in tandem with other proinflammatory cytokines. These responses were indicative of activated cytotoxic T cell immunity, which contributed to the control of MNV infection in vivo [58,59]. 

Human T cell epitopes have recently been identified [81]. A total of three GII.4-1999 T cell epitopes were proposed as immunodominant. However, only a single epitope was fully characterized, as it exhibited cross-reactivity in more than one patient. HLA-A2, the most common MHC allele family in North Americans [186,187], binds GII.4 amino acids 139 to 148 in the S domain of VP1. Residues 140 and 148 were determined to be anchoring positions for HLA-A 02:01. IFN-γ, IL-2, and TNF-α levels increased when polyfunctional CD3+ CD8+ T cells bound this peptide [188]. Sequence alignments revealed that this 10-amino acid region was highly conserved in VP1 across both GI and GII strains and remained unchanged in the GII.4 genotype until the emergence of the Sydney 2012 strain. Up to three amino acid substitutions could be tolerated without reducing the binding affinity to HLA-A [188], thus serving as evidence of a highly potent, cross-reactive T cell epitope. The importance of this epitope requires further investigation, but the identification of this epitope within the S domain indicates a possible limitation for P particle vaccines that lack the shell [177,188]. Further characterization of novel T cell epitopes, within VP1 and other viral proteins, should enhance efforts to develop cross-protective HuNoV vaccines and possibly, norovirus-specific T cell therapies.

## 8. Applications of Norovirus Epitope Studies

### 8.1. Improved Human Norovirus Diagnostics

Today, most norovirus diagnostic assays are nucleic acid-based with multiplex panels and genome sequencing [189]. However, rapid and sensitive point-of-care tests that utilize cross-reactive mAbs can play an important role in settings where nucleic acid testing is impractical. There is currently one FDA-approved test for HuNoV that is based on an immunochromatographic assay, which shows high sensitivity for GII genotypes, and GII.4 in particular [190,191,192]. For most outbreak situations, this test would be acceptable, as GII.4 is the predominant genotype in circulation. However, the lower sensitivity for GI genotypes could hamper diagnosis of sporadic GI cases [190]. The continued improvement and availability of point-of-care tests would facilitate epidemiological monitoring of uncommon HuNoV genotypes and aid in proper patient diagnosis and treatment [193,194]. Overall, these efforts would decrease the number of infections and cost of care per infection.

Several IgGs and scFvs have been generated that may expand the possible genotypes that can be detected in immunoassays, including NS14, NS22, and HJT-R3-A9 (Appendix A). In fact, a combination of NS14 and HJT-R3-A9 was used to create a diagnostic platform that detected 25 genotypes of HuNoV in patient stool samples [134]. Nano-85 has also been used to create a nanobody-based lateral flow immunoassay (Nano-IC), which recognizes both outbreak genotypes GII.4 and GII.17 [128]. Further, novel diagnostic assays such as an HuNoV bioluminescence enzyme immunoassay (BLEIA) have been developed. BLEIA is a fully automated, ultrasensitive assay that has demonstrated broad cross-genogroup reactivity with a rate of completing 120 tests/hour [195]. Additionally, an external force-assisted near-field illumination biosensor (EFA-NI biosensor) was created for HuNoV. Coated with cross-reactive mAbs, the EFA-NI biosensor recognized as few as 40 viral particles [196], suggesting that these rapid, ultra-sensitive immunoassays may complement standard nucleic acid-based methods in clinical settings.

### 8.2. Therapeutic Potential

Norovirus mAbs have promise for use in immunoprophylaxis and treatment. Passive antibody therapy with HuNoV mAbs could provide immediate protection for individuals at risk for infection, such as transplant recipients and those exposed during outbreaks [9,197]. Therapeutic antibodies could be used to clear chronic norovirus infection in immunocompromised individuals. However, thus far, it has been difficult to identify and isolate potent cross-protective mAbs. To illustrate this point, over 100 mAbs derived from primed mice were characterized before the cross-blockade mAb GII.4G was identified [37]. Further characterization of essential and conserved epitopes is needed before mAb therapeutics can be widely implemented. Alternatively, cocktails of genotype-specific protective mAbs might provide a more immediate option.

The therapeutic potential of the current repertoire of HuNoV mAbs is variable, and some would require humanization by genetic engineering. Protective HuNoV mAbs produced in non-primate animals, including mice and chickens, cannot be directly administered to humans due to allergy or hypersensitivity concerns [198,199,200]. Chimpanzee mAbs are virtually identical to human mAbs and demonstrate effective, long-lasting half-lives for the treatment of viral pathogens [201,202]. Although a number of chimpanzee mAbs are available for HuNoV, further testing in these animals [87] for efficacy is unlikely. Human mAbs offer excellent therapeutic potential, as PBMC isolation and expression of naturally-occurring, matched heavy- and light-chains bypasses the need for humanization altogether [36,118,124]. Immunoglobulin replacement therapy in the form of pooled, cloned antibodies from healthy individuals within a community or geographical region might also allow passive transfer of “herd immunity” to local at-risk patients. 

Therapeutics based on scFv and VHH circumvent the need for humanization of Fc domains. Moreover, VHH coding sequences can be engineered into any human immunoglobulin scaffold [203,204], and scFvs can be engineered into multimeric forms to be more stable and recognize several targets [205]. The small size of both scFvs and VHHs would allow buried epitopes to be more accessible. VHH and scFv size would also not interfere with the patient’s natural humoral response to HuNoV infection or vaccination. However, to achieve the same level of protection as standard mAbs, an estimated 20 times more VHH or scFv would be required per administration [102]. VHH yields are high in several biotechnical platforms, including yeast and bacteria [206,207], which could offset costs and production times. Administration to the gut may be limited for scFvs, as exposure to gut proteases causes partial instability only alleviated by scaffolding to another platform [206]. VHHs, on the other hand, may survive an oral route of administration due to their high re-folding efficiency, high solubility, and resistance to proteases and chemical denaturants [208,209].

Administration of pooled immunoglobulins from healthy donors has been shown to clear chronic infections in some immunocompromised patients [210,211,212]. However, the success of commercially-available IVIG is variable. One option for enhancing the efficacy of IVIG may be co-administration of HuNoV antibodies and human milk oligosaccharides (HMOs) by the oral route. Interestingly, recent studies showed synergistic and additive effects of blocking HBGA binding when VLPs were treated with nanobodies supplemented with the soluble HMO, 2’-fucosyllactose (2’FL) [130]. HMOs have been shown to competitively bind the HBGA pocket of HuNoV, possibly mimicking HBGAs and acting as receptor decoys [213,214,215]. The positive effect of adding 2’FL was not limited to a single mode of action or single HuNoV genogroup, supporting the cross-protective potential of this treatment alongside mAb administration. Other combinations of therapies, including those that enhance or replace missing cellular immunity, hold promise for clearing chronic HuNoV infection.

### 8.3. Design of Universal Vaccines

Early adult challenge studies reported that norovirus immunity was predominantly homologous and lasted from six months to two years post-infection [4,84,216,217,218]. Assay limitations and overwhelming challenge doses at the time may have underestimated the duration of immunity [76,122]. The epochal evolution of GII.4, the modeling of HuNoV incidence, and mutations in immunodominant epitopes instead argue that protective immunity lasts closer to four to nine years [32,36,219,220]. Similarly, mucosal IgA and T cell responses in norovirus-challenged patients strongly support the existence of long-term protective immunity [78]. The breadth of this immunity across genotypes continues to be characterized. A recent report analyzed all available data from birth cohort studies [43]. Patterns of high reinfection rates in children provided evidence that cross-protection was observable within related clusters of genotypes provisionally called “immunotypes” [43]. Continued epitope mapping may be important in understanding the nature of these antigenic relationships.

An effective HuNoV vaccine would ideally induce long-lasting protection against a wide range of genotypes. Polyclonal cross-blockade antibody responses can be observed following natural infection and artificial challenge, even against highly variant GII.4 strains [38,78,121,147,180,181]. This suggests that there are targetable, shared epitopes across norovirus genotypes, though this observation may have been confounded by unknown pre-exposure histories of patients enrolled in these studies. The identification of conserved blockade epitopes buried within the VLP structure may call for VLP vaccines to be tailored to expose these often-inaccessible regions. Treatment with nanobodies that force this exposed conformation have been proposed, as nanobody-facilitated open conformations were achieved for picornaviruses [221]. Alternatively, VLPs could be engineered to have decreased temperature-related flexibility, thereby enhancing antigen presentation and improving the shelf life of VLP-based vaccines [222,223].

Currently, numerous monovalent and multivalent vaccines against HuNoV have been developed, with several progressing into phase I/II clinical trials [82,86,105,147,181,224,225,226,227,228,229,230]. These VLP vaccines elicit strong IgA and IgG responses and expansion of antibody-secreting cells [82,228,229,230,231]. Ideal multivalent vaccine candidates will likely need to include GI and GII immunogens, as broad blockade has been observed in mice inoculated with multiple HuNoV VLPs, even against strains not included in the cocktail [105,181,224,225]. However, vaccination may not be the direct cause of this cross-genogroup protection. Rather, vaccination may have recalled previous GII-specific memory B cells and matured antibody populations through additional selection [86]. This recall of past HuNoV exposures has been recorded for several vaccine and challenge studies [86,121,227], illustrating the importance of natural history in the development of protective immunity.

Primary infections with HuNoV were initially thought to cause “original antigenic sin” (OAS). This theory asserts that an individual’s first HuNoV strain exposure would predetermine dominant immune responses to subsequent HuNoV infections, and evidence for such an effect has been observed in a young child [232]. However, “antigenic seniority” may be the better descriptor of norovirus anamnestic responses. Antigenic seniority states that preexisting immunity, though governed by primary HuNoV exposure, does not prevent the ability to generate new blockade antibodies and T cell responses [86,233]. To support this theory, it has been suggested that there is no genetic restriction in the ability to make new blocking antibodies to HuNoV [124]. Additionally, an in vitro neutralization study provided evidence that the HBGA blockade response was not restricted to a specific genetic sequence motif in antibody repertoires [118]. However, antibody titers tended to plateau following vaccine administration, which resulted in lower fold increases in antibody production [86,147]. If memory antibody titers are high prior to vaccination, this leaves little opportunity for new blockade antibodies to be produced. This phenomenon has been observed for measles and influenza A [234,235] and argues for vaccination early in life to establish a broad memory response.

Alternatively, this memory response can work with vaccination to induce cross-protective responses against highly variable, immunodominant epitopes. Though the antigenic variation of immunodominant epitopes is high, a vaccination study noted evidence of polyclonal serum antibodies possibly containing a rare class of cross-reactive Epitope A-binding antibodies [86]. By repeatedly vaccinating against engineered VLPs that present unique Epitope A sequences, B cells that recognize conserved residues on or near this immunodominant epitope were selected to establish broadly effective, Epitope A-mediated protection [105]. This ability to harness preferential antigen presentation on VLP vaccines could therefore selectively drive immune responses away from hypervariable epitopes and toward conserved, cross-protective antigenic sites [160]. Studies of antibody repertoires pre- and post-vaccination, coupled with structural studies of mAb-bound VLPs, would continue to inform how HuNoV humoral immunity evolves over time and can be tailored towards long-lasting and cross-reactive protection.

Current vaccination efforts report a reduction in disease severity, but varied degrees of protection from reinfection [86,87,147,227]. Achieving increases in titer and cross-blockade activity has been the measurement of successful HuNoV vaccine candidates. However, many viral infections are not contained by antibody responses alone, and studies with MNV in mice have demonstrated that T cell immunity is critical for viral clearance [54,58]. In certain immunocompromised patients with chronic norovirus infection, with varying degrees of humoral immunity intact, the virus was cleared only following the re-acquisition of functional T cells [236,237]. Additionally, one study observed that children developed cellular immune responses to HuNoV infections independently of seroconversion and proposed that serial exposure may develop a stable memory T cell response [80]. Vaccination early in life could utilize this preference toward T cell immunity to establish longer-lasting protection.

Several vaccination studies have measured evidence of T cell stimulation through the increased levels of IFN- γ, IL-4, and activated CD4+ and CD8+ T cells [76,82,231,238]. Additionally, there is evidence of T cell memory and cross-stimulation against closely related strains in naturally infected patients [5,58,76,78,81]. Stimulation of T cells with a highly conserved T cell epitope induced a memory response in cells isolated from control patients, providing evidence that inclusion of these conserved epitopes in vaccines might stimulate this cellular memory [81]. How VLP-based vaccination compares to natural infection in the stimulation of T cell immunity remains unclear, as VLP vaccines represent only one viral protein. Adoptive T cell therapy from mice immunized with VLPs failed to protect recipient mice against live virus challenge, suggesting that other proteins might bear more potent T cell epitopes [224]. The inclusion of potent T cell epitopes in vaccines may complement cross-protective humoral responses and establish better viral clearance, enhanced memory responses, and increased protection against reinfection.

## 9. Summary

Decades of humoral immune studies have addressed the complex antigenic diversity of the noroviruses. Monoclonal antibodies have played a key role in defining regions of interest in the major capsid protein that may be therapeutically targetable. It is now confirmed that certain HuNoV mAbs mapping to the P domain of VP1 have both HBGA blockade and neutralization activity, an important validation of the surrogate carbohydrate blocking assays. These functional epitopes are consistently conformation-dependent, an important structural feature to consider in protein engineering or reverse engineering of vaccine antigens. Insight has been gained into mechanisms by which antibodies neutralize beyond blocking attachment to cells, including particle aggregation or disassembly. Additionally, the discovery of temperature-sensitive epitopes has provided insight into the dynamic nature of norovirus capsids, a structural feature that could potentially be harnessed to gain access to hidden functional epitopes. Such knowledge will be helpful in the design of antibodies to treat or prevent norovirus infection, and in profiling the humoral immune response to candidate vaccines. Importantly, an understanding of how antibodies function can inform the selective immune pressures that drive antigenic diversity and escape from herd immunity.

Epitope mapping has now established that there are conserved B and T cell epitopes among noroviruses, despite the genetic diversity. The role of these cross-reactive epitopes in protective immunity is not yet known. Interestingly, both conserved B and T cell epitopes map to the S and C-terminal P1 domain. While B cells and the antibodies they secrete contribute to protection and control of norovirus infection, the role of T cells should also be considered in achieving long-lasting and cross-genotypic protection. Given the relative abundance of antibody mapping studies, further mapping of functional T cell epitopes is needed. The stimulation of lasting immunity at the intestinal mucosal surfaces may prove challenging for the noroviruses. Knowledge of norovirus B and T cell epitopes will undoubtedly continue to play an important role in meeting this challenge.

## Figures and Tables

**Figure 1 viruses-11-00432-f001:**
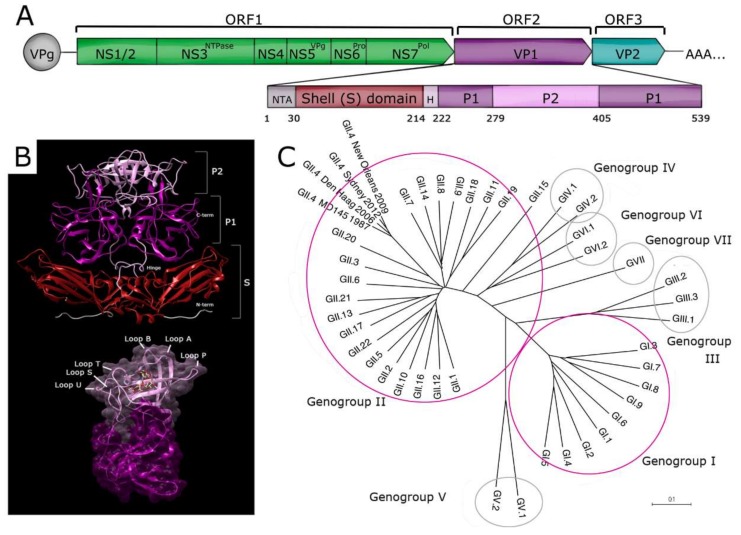
(**A**) Organization of the human norovirus genome. ORF1 (green) encodes the nonstructural proteins, ORF2 (purple) encodes the major structural capsid protein, VP1, and ORF3 (blue), encodes the minor structural protein, VP2. Amino acids are numbered according to a representative GI.1 genome (GenBank: KF429765.1). The VP1 protein is divided into two major domains: Shell (S) and Protruding (P) [17]. The S domain is immediately preceded by a short N-terminal arm (NTA), and the S and P domains are connected by a flexible hinge region (H). The P domain is further subdivided into P1 and P2. (**B**) Structural modeling of the capsid VP1 dimer. VP1 dimerizes from two protein chains (A and B), with the S domain constructing the inner part of the capsid that surrounds the RNA genome. P1 domains fold into each other, allowing P2 to become the most exposed portion of the dimer. Loops are labeled according to GI.1 nomenclature [18]. ChimeraX was used to model loops and HBGA binding (GI.1 PDB 2ZL6). (**C**) Genetic diversity of norovirus VP1. Representative VP1 sequences from each genotype were aligned using MEGA. Genotypes cluster into Genogroups GI through GVII. The majority of human noroviruses belong to Genogroups GI and GII.

**Figure 2 viruses-11-00432-f002:**
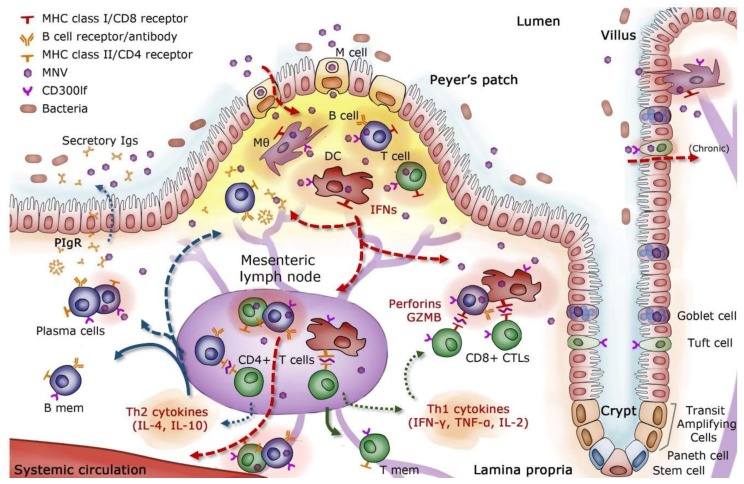
– Overview of proposed adaptive immune responses to murine norovirus infection in vivo. Movement of virus or virus-infected cells is marked by red dashed lines with arrows. B cell- (blue) and T cell- (green) related movement/cytokine or immunoglobulin secretion are marked by blue and green dashed lines, respectively. Maturation of memory B and T cells is marked by solid lines. Acute MNV infection begins with endocytosis at M cells found in Peyer’s patches of the intestinal tract [47]. Following entry into the lamina propria, MNV infects CD300lf-positive macrophages (Mθ), dendritic cells (DCs), B cells, and T cells [46,48,49,50]. Innate immune responses are mediated through the production and release of interferons (IFNs), which aid in the activation of the adaptive immune response. Professional antigen-presenting cells (APCs) can recognize antibody-tagged virus via Fc receptors as part of phagocytosis. Additionally, APCs can migrate to mesenteric lymph nodes (MLNs). Presentation of MNV peptides on MHC class I molecules leads to the stimulation of Th1 proinflammatory responses. CD4+ Th1 cells release various cytokines that upregulate the activity of CD8+ cytotoxic T lymphocytes (CTLs). These T cells interact with MNV peptides presented on MHC class I molecules on infected cells, which initiates the release of cytotoxic molecules such as perforins and granzyme B (GZMB) [54,58]. APCs that have migrated to MLNs can also present MNV antigens on MHC class II molecules and elicit an upregulation of Th2 responses, which help mature B cells. Following T cell-mediated maturation, B cells migrate to sites of active infection and begin secreting large amounts of immunoglobulins (Igs) as plasma cells. Ig secretion is mediated by the polymeric immunoglobulin receptor (pIgR) on enterocytes. The majority of MNV replication and adaptive immune responses to MNV take place in the intestinal mucosa, though MNV has been reported in more systemic organs like the spleen [49]. MNV has rarely been found in villi, which could be the result of minimal replication in intraepithelial lymphocytes (IELs) [46]. Certain strains of MNV can become persistent and appear to target tuft cells, specialized epithelial cells that express the CD300lf viral receptor and thought to be immune-privileged [62,63].

**Figure 3 viruses-11-00432-f003:**
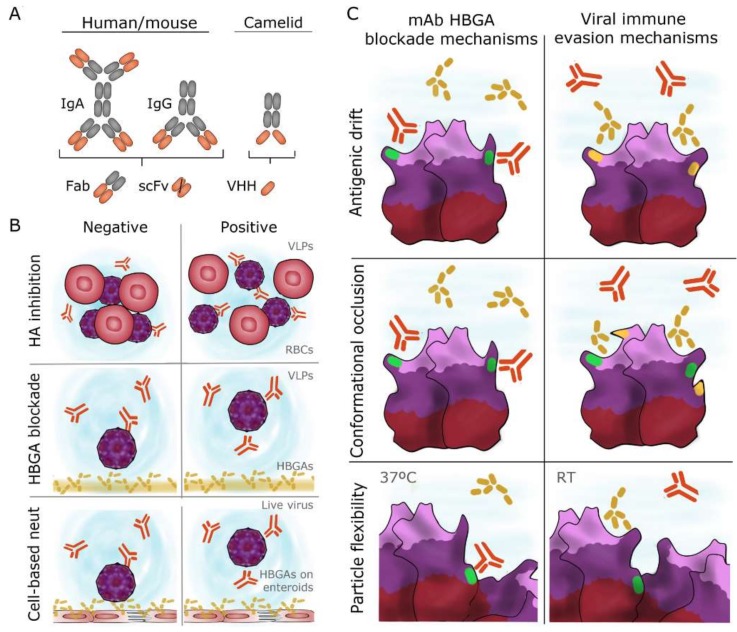
Monoclonal antibody types and their functional characterization. (**A**) Types of monoclonal antibodies used to map norovirus epitopes are shown. Intact immunoglobulins derived from traditional hybridoma technology have been used extensively. Various forms of Fab fragments can be isolated from Fc regions through peptide digestion of intact immunoglobulins. Recombinant antibody technology based on the cloning and engineering of selected complementarity-determining regions (CDRs) in various immunoglobulin scaffolds have gained increasing use. Single-chain variable fragments (scFvs) are created by expression of the single heavy and light chain sequences of a complete antigen-binding site joined by a flexible linker. Variable heavy chain (VHH) antibodies isolated from camelids have proven useful for mapping norovirus epitopes due to their small size and accessibility to hidden epitopes. (**B**) Commonly used assays to assess antibody function are shown. Surrogate neutralization assays (HAI and HBGA blockade) measure the ability of antibodies to interfere with the binding of HuNoV VLPs to blood group carbohydrates. Positive HAI activity is scored by the ability of antibodies to prevent blood cell agglutination by HuNoV VLPs. Blockade activity measures the capacity of antibodies to inhibit the binding of VLPs to immobilized HBGA carbohydrate ligands. Neutralization assays are available for MNV, and now, HuNoV in a newly established enteroid cell culture system. Preincubation of live virus with neutralizing antibody results in a decrease of viral infection in vitro, measured by decreased viral titer or RNA genome copies. (**C**) Mechanisms of mAb-mediated HBGA blockade and corresponding viral immune evasion strategies. mAb binding epitopes are colored green on the illustration of dimerized VP1. In C1 and C2, mAbs can bind directly to the HBGA binding pocket or to regions immediately adjacent to the HBGA binding pocket. This induces blockade through competitive binding and/or steric hindrance. Additionally, mAbs can bind distant epitopes, which influence the structure of the P domain and force the HBGA binding pocket into an unfavorable position. Antigenic drift combats these mechanisms of HBGA blockade by changing either the residues directly involved in mAb recognition (C1) or residues adjacent to mAb binding epitopes (C2) (depicted in orange). In the latter case, amino acid changes can result in different particle or P domain conformations, which can occlude mAb binding as well as alter HBGA binding profiles altogether (C2). Epitopes buried within the VLP structure can also influence the structure of HBGA binding domains when exposed at temperatures higher than RT (37 °C) (C3). The temperature-sensitivity of particle conformation can hide these buried epitopes at lower temperatures, thereby preventing mAb recognition and subsequent HBGA blockade.

**Figure 4 viruses-11-00432-f004:**
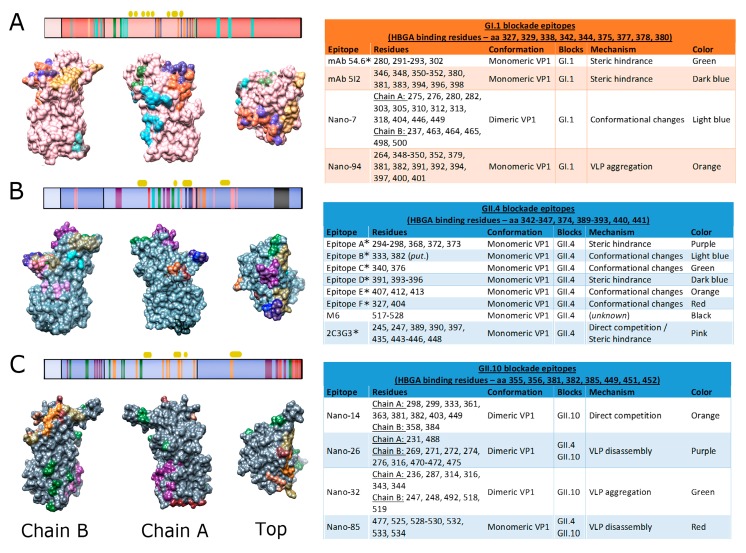
Comparison of HBGA blockade epitopes mapped to (**A**) GI.1, (**B**) GII.4, and (**C**) GII.10 P domains. HBGA binding residues are denoted in gold above the linear representation and the three-dimensional models of HuNoV P domains. Adjacent tables record epitope specificities and coloring that corresponds to the positions of the amino acids on both the linear and three-dimensional diagrams. Antibodies marked with an asterisk (*) do not have an antibody/virus structure associated with their epitope definition. ChimeraX was used to model amino acid binding sites (GI.1 PDB 2ZL6, GII.4 PDB 2OBS, GII.10 PDB 3ONU).

**Figure 5 viruses-11-00432-f005:**
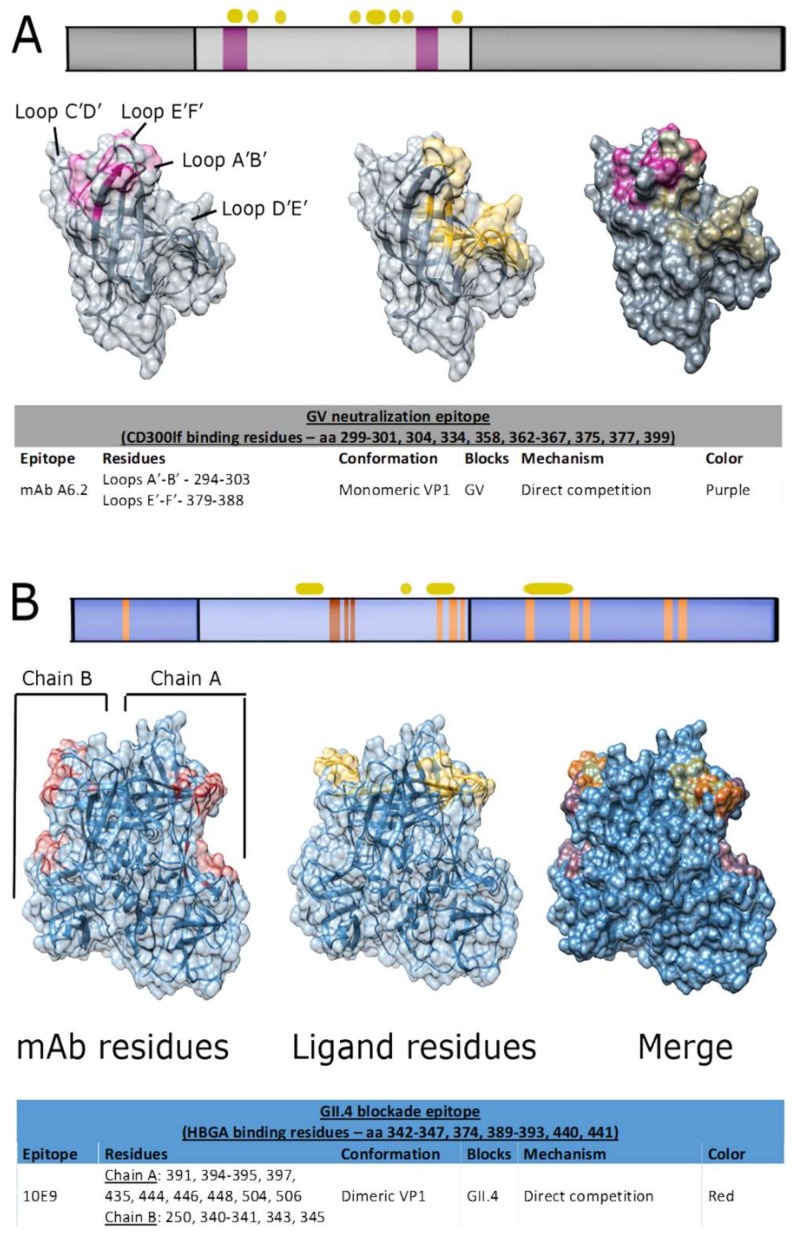
Relationship of two neutralizing B cell epitopes to receptor or HBGA binding sites. (**A**) MNV neutralizing epitope mapping to the P2 domain. Neutralizing mAb A6.2 binds to a conformational epitope (purple) that overlaps the CD300lf receptor binding domain (gold). (**B**) GII.4 neutralizing epitope mapping to the P2 domain. Neutralizing mAb 10E9 binds to an epitope that spans the P2 strains [65]. 2D3 is predicted to recognize some of the same residues as MAb A6.2 and possibly overlap with the CD300lf binding site [172]. Interestingly, resistance mutations to 2D3 took about 20 passages to become detectable, which suggested that the 2D3 mAb recognized critical residues involved in receptor binding [65]. Amino acid changes that resulted in loss of neutralization for both MAb A6.2 and 2D3 forced the P domain into an unfavorable conformation, instead of altering the dimer interface (orange). HBGA binding residues are highlighted in gold. Adjacent tables record epitope specificities and coloring that corresponds to the positions of the amino acids on both the linear and three-dimensional diagrams. ChimeraX was used to model amino acid binding sites (MNV PDB 6C6Q, GII.4 PDB 6EWB).

**Figure 6 viruses-11-00432-f006:**
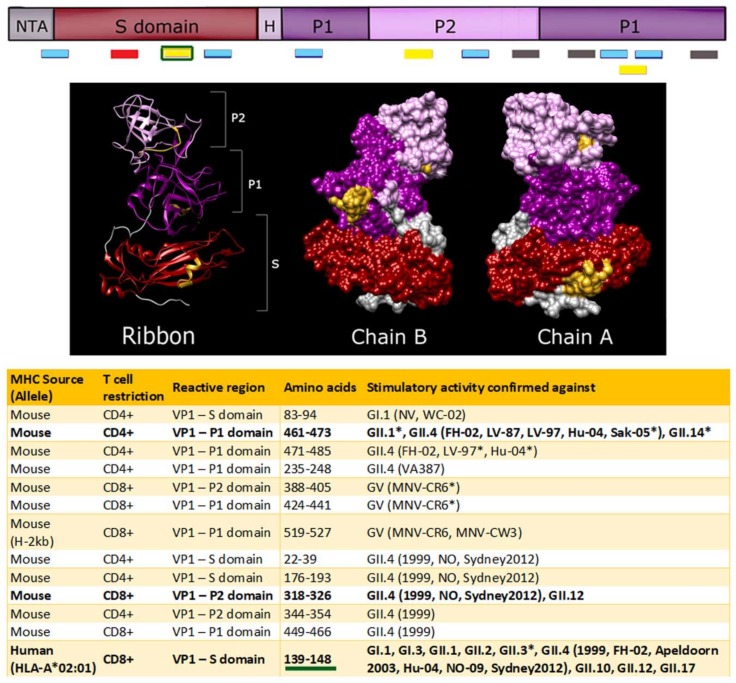
Summary of T cell epitopes mapping to the norovirus VP1 sequence. T cells were isolated from mice and humans with either prior norovirus infection or immunization with VLPs. T cells were stimulated with peptides spanning a norovirus genotype of interest. Reactive epitopes are recorded in the table and illustrated on the linear schematic of VP1. MNV-specific epitopes are shaded grey, GI genotype-specific epitopes are shaded red, and GII genotype-specific epitopes are shaded blue. Three cross-genotype reactive epitopes (bolded in the table), including a human-derived T cell epitope mapping in the S domain (outlined in green), are shaded in gold and modeled on the 3D structure of GI.1 NV VP1. Taken together, the majority of T cell epitopes have been mapped to the S and P1 domains of VP1.

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
