# Peer review of "The Antigenic Topology of Norovirus as Defined by B and T Cell Epitope Mapping: Implications for Universal Vaccines and Therapeutics"

_viruses, 2019, doi:10.3390/v11050432_

Reviewer 1 Report

This was an extraordinarily useful and well-written review manuscript; I imagine it will become a key resource for anyone even peripherally interested in adaptive responses to noroviruses! It was also impressively clear, well-organized, and nicely-worded - kudos to the authors for putting together such an excellent review of the field!

I have only very minor comments/suggestions:

There are a few portions of figures where the text is quite small and difficult to read. In Fig. 1C, if font size could be increased a bit for the individual virus designations that would be helpful. Similarly, the text in the tables in Figures 4/5 is a bit of a strain to read - would it be possible to move the tables to the side of the structures to make more room to increase font?

In Figure 2, the dotted vs dashed arrows were challenging to make out; perhaps adding in additional colors or some other means of designating different cell actions would resolve this? Also Figure 2 legend could use citations. Finally, the expression of CD300lf on tuft cells is almost exclusively towards the lumen, and persistent infection is likely independent of M cell-mediated entry into the lamina propria based on recent publication (Rag2/Il2rg mice can get infected with CR6, PMID 30936486), so figure should be adjusted to reflect this.

In text (around lines 145-147), would be appropriate to mention that tuft cells seem to be an immune-privileged niche from T cell responses and cite related work (PMID 29031786).

In lines 162-164, also appropriate to mention B cell model for HNoV cultivation (PMID 26513671), and also to add reference for in vivo HNoV replication in enterocytes.

In Fig. 6, would it be possible to provide a similar comparison of the T cell epitopes by mapping onto structures as was provided for B cell epitopes? Or alternately, just add on T cell epitopes on top of prior structures from Figures 4/5 for direct comparisons. The authors make the point (lines 857) that conserved epitopes map to overlapping domains and it would be helpful to visualize that more directly.

Author Response

Reviewer 1

 Comments and Suggestions for Authors

This was an extraordinarily useful and well-written review manuscript; I imagine it will become a key resource for anyone even peripherally interested in adaptive responses to noroviruses! It was also impressively clear, well-organized, and nicely-worded - kudos to the authors for putting together such an excellent review of the field!

Response:  Thank you for the positive feedback.  

Reviewer:  There are a few portions of figures where the text is quite small and difficult to read. In Fig. 1C, if font size could be increased a bit for the individual virus designations that would be helpful. Similarly, the text in the tables in Figures 4/5 is a bit of a strain to read would it be possible to move the tables to the side of the structures to make more room to increase font?

Response:  The font size for Figure 1C has been increased, and the table in Figure 4 has been moved to the side of the structures. Tables in Figure 5 have been enlarged.

Reviewer:  In Figure 2, the dotted vs dashed arrows were challenging to make out; perhaps adding in additional colors or some other means of designating different cell actions would resolve this? Also Figure 2 legend could use citations.

Response:  The difference between dashed and dotted was eliminated, and the color scheme of B vs. T cell responses are explained in the beginning of the figure legend.  Citations have been added.

Reviewer:  Finally, the expression of CD300lf on tuft cells is almost exclusively towards the lumen, and persistent infection is likely independent of M cell-mediated entry into the lamina propria based on recent publication (Rag2/Il2rg mice can get infected with CR6, PMID 30936486), so figure should be adjusted to reflect this.

Response:  The apical expression of CD300lf on tuft cells has been added to Figure 2.

Reviewer:  In text (around lines 145-147), would be appropriate to mention that tuft cells seem to be an immune-privileged niche from T cell responses and cite related work (PMID 29031786).

Response: This reference has been added and the immune privilege noted.  

Reviewer:  In lines 162-164, also appropriate to mention B cell model for HNoV cultivation (PMID 26513671), and also to add reference for in vivo HNoV replication in enterocytes.

Response:  We added the Jones et al. reference (50) about the B cell model at the beginning of the paragraph when discussing similarities of MNV and HuNoV. The Karandikar reference (69) was cited as evidence of in vivoHuNoV replication in enterocytes.

In Fig. 6, would it be possible to provide a similar comparison of the T cell epitopes by mapping onto structures as was provided for B cell epitopes? Or alternately, just add on T cell epitopes on top of prior structures from Figures 4/5 for direct comparisons. The authors make the point (lines 857) that conserved epitopes map to overlapping domains and it would be helpful to visualize that more directly.

Response:  A figure modeling the mapped cross-genotypic epitopes has been added.

Reviewer 2 Report

This is a well written review that is rather expansive in the number of studies included. I do, however, have a few suggestions that I think will greatly improve its accuracy.

The first point is not entirely the fault of the authors, but rather the traditional use of some common terms. The definition of ‘epitope’ is; “A molecular region on the surface of an antigen capable of eliciting an immune response and of combining with the specific antibody produced by such a response.” My problem is that the review lumps together a wide range of different studies, with divergent methodologies, under the assumption that all mutations that affect antibody binding define an epitope. Under many circumstances, this is not necessarily a bad assumption. However, as noted in a number of the cited papers, there is a growing database of virus/antibody structures where the contact residues have been identified. While not all of these contact residues necessarily block antibody binding upon mutation, the antibody binding interface closely fits the definition of an epitope. A structure-based definition of an epitope is important for several reasons. 1) If one hopes to recapitulate an antigenic site to create a vaccine, then this is the surface one needs to present to the immune system. 2) Perhaps most importantly, a number of cited papers discuss the fact that some of the escape mutations are far away from the actual antibody binding contact or buried in the P domain. These residues are therefore not part of an ‘epitope’ in the traditional sense and, on their own, cannot be assumed to play a role in antibody binding. The review tends to gloss over this important delineation in several places. I really think that this needs to be discussed – that mutations that affect antibody binding need not be directly involved in antibody binding and these are not ‘epitopes’, per se, but rather residues that can affect the overall structure of the protein that, in turn, can affect epitope structures. Therefore, some of the figures where the ‘epitopes’ are mapped onto the capsid structures need to be given the caveat that if there is not a structure of the capsid/antibody complex, then there is a possibility that the antibody is not actually binding to these locations. I really think the field could benefit from a more nuanced discussion of epitopes.

A second point where the authors could help in clarifying the field is with regard to mechanisms of neutralization. For example, in the case of HBGA binding human noroviruses, it should be noted that antibodies can block cell attachment without directly binding to the sugar binding site on the virus. Antibodies, even IgG’s, are huge compared to the virus – equal to the radius of the particle. Therefore, even a very few antibodies binding to a viral particle will most certainly block viruses from getting close enough to the membrane for attachment. This was suggested to be an important consideration in the case of rhinovirus and then latter with HIV. So, yes, the term ‘blocking’ to describe the effects of antibodies on the agglutination assay is fine, but I, again, think that caveats are needed to remind the reader that this binding abrogation can be due to the massive bulk of antibodies and not necessarily due to overlapping carbohydrate/antibody binding interactions. The introduction figure 2 is wonderful but could be enhanced by noting that the Fc receptors on macrophages, neutrophils, dendritic cells are looking for virus/antibody complexes and that that ‘tagging’ of the viruses for phagocytosis could be even more important than steric blockade of virus/receptor interactions.

Some of my more specific comments.

1)    Lines 255-266. I would be a bit careful in suggesting that an IgA differs from an IgG with regard to bulk. The only way to demonstrate this is if you have an IgG and IgA with exactly the same paratope. Further, one would have to demonstrate that a dimeric IgA is more effective just because of its bulk and not because of potential aggregation of the virions. Again, it seems to suggest a lesser role of paratope/epitope interaction compared to antibody isotype.

2)    Lines 453-457. I would word this section rather differently. Antibodies do not cause conformational changes, per se, but rather select for conformations that present the binding sites. I would not at all say that the X-ray structure of G1.1 demonstrates inflexibility of the capsid such that the antibody cannot bind. Indeed, this structure was done a very high salt concentration and at low pH. It represents the conformation at a particular condition. Further, we know from other studies on three other genotypes that the particles are indeed flexible at more physiological conditions.

3)    Lines 567-569. It is not clear what is meant by the epitope not being clearly defined. This goes back to the start of my comments. The escape mutations here are not, in the strictest sense, part of the epitope. The Ab/virus contacts are clearly defined by the cryo-EM structure and none of those residues were able to mutate to block antibody binding. This, again, shows the problem of defining an epitope by residues that affect antibody binding. I am not sure the best way to fix this other than to note at the start that epitope is being defined as residues that block antibody binding – but that these may include residues that do not directly contact the antibody but can cause conformational changes that do. In the absence of structural information on particular antibody/virus complexes, one cannot say for certain if those residues are part of the true definition of an epitope.

Author Response

Reviewer 2

This is a well written review that is rather expansive in the number of studies included. I do, however, have a few suggestions that I think will greatly improve its accuracy.

The first point is not entirely the fault of the authors, but rather the traditional use of some common terms. The definition of ‘epitope’ is; “A molecular region on the surface of an antigen capable of eliciting an immune response and of combining with the specific antibody produced by such a response.” My problem is that the review lumps together a wide range of different studies, with divergent methodologies, under the assumption that all mutations that affect antibody binding define an epitope. Under many circumstances, this is not necessarily a bad assumption. However, as noted in a number of the cited papers, there is a growing database of virus/antibody structures where the contact residues have been identified. While not all of these contact residues necessarily block antibody binding upon mutation, the antibody binding interface closely fits the definition of an epitope. A structure-based definition of an epitope is important for several reasons. 1) If one hopes to recapitulate an antigenic site to create a vaccine, then this is the surface one needs to present to the immune system. 2) Perhaps most importantly, a number of cited papers discuss the fact that some of the escape mutations are far away from the actual antibody binding contact or buried in the P domain. These residues are therefore not part of an ‘epitope’ in the traditional sense and, on their own, cannot be assumed to play a role in antibody binding. The review tends to gloss over this important delineation in several places. I really think that this needs to be discussed – that mutations that affect antibody binding need not be directly involved in antibody binding and these are not ‘epitopes’, per se, but rather residues that can affect the overall structure of the protein that, in turn, can affect epitope structures. Therefore, some of the figures where the ‘epitopes’ are mapped onto the capsid structures need to be given the caveat that if there is not a structure of the capsid/antibody complex, then there is a possibility that the antibody is not actually binding to these locations. I really think the field could benefit from a more nuanced discussion of epitopes.

Response:  We appreciate the reviewer’s concerns on terminology.  We have addressed these by the following modifications of the manuscript: 

1.    We have clarified our use of the term “epitope” beginning on line 230 of the revised manuscript by adding the following text:  “It should be noted that throughout this review, an “epitope” is considered defined by the continuous or non-continuous residues of VP1 that interact directly with the mAb. Mapping data that includes structural analyses of the antibody and viral antigen complex are required for optimal precision. However, mapping studies may report residues that indirectly affect the presentation and recognition of an epitope, and these published data are included in the tables.”

2.    Mutations causing an indirect disruption of mAb/virus interactions, as described by the reviewer, have been denoted in the supplementary tables as “influenced by (residue, number).”

3.    An asterisk (*) has been added to the antibodies of Figure 4 to denote the epitopes which have not been determined using structural studies. 

Reviewer:  A second point where the authors could help in clarifying the field is with regard to mechanisms of neutralization. For example, in the case of HBGA binding human noroviruses, it should be noted that antibodies can block cell attachment without directly binding to the sugar binding site on the virus. Antibodies, even IgG’s, are huge compared to the virus – equal to the radius of the particle. Therefore, even a very few antibodies binding to a viral particle will most certainly block viruses from getting close enough to the membrane for attachment. This was suggested to be an important consideration in the case of rhinovirus and then latter with HIV. So, yes, the term ‘blocking’ to describe the effects of antibodies on the agglutination assay is fine, but I, again, think that caveats are needed to remind the reader that this binding abrogation can be due to the massive bulk of antibodies and not necessarily due to overlapping carbohydrate/antibody binding interactions. 

Response:  We agree with the reviewer. For clarification, we edited written the following statement to read:  

“Monoclonal antibodies that bind residues immediately adjacent to an HBGA binding site could prevent ligand interaction via steric hindrance by the bulky antibody itself.”

Reviewer:  The introduction figure 2 is wonderful but could be enhanced by noting that theFc receptors on macrophages, neutrophils, dendritic cells are looking for virus/antibody complexes and that ‘tagging’ of the viruses for phagocytosis could be even more important than steric blockade of virus/receptor interactions.

Response:  Thank you for noting this excellent point.  We have added this concept to the Figure 2 legend.

Some of my more specific comments.

1)    Lines 255-266. I would be a bit careful in suggesting that an IgA differs from an IgG with regard to bulk. The only way to demonstrate this is if you have an IgG and IgA with exactly the same paratope. Further, one would have to demonstrate that a dimeric IgA is more effective just because of its bulk and not because of potential aggregation of the virions. Again, it seems to suggest a lesser role of paratope/epitope interaction compared to antibody isotype. 

Response:  It is true that the paper we cited for this concept did not directly verify the relationship between paratope and isotype. To insert a note of caution on this point, we have modified the text as follows:  

“Based on studies with the human monoclonal IgA, mAb 5I2, the authors proposedthat dimeric IgA blockade activity may havestemmed from its large molecular weight and steric hindrance of VLP binding to carbohydrates [125].”

2)    Lines 453-457. I would word this section rather differently.  Antibodies do not cause conformational changes, per se, but rather select for conformations that present the binding sites. I would not at all say that the X-ray structure of G1.1 demonstrates inflexibility of the capsid such that the antibody cannot bind. Indeed, this structure was done a very high salt concentration and at low pH. It represents the conformation at a particular condition. Further, we know from other studies on three other genotypes that the particles are indeed flexible at more physiological conditions.

Response:  We agree with the reviewer. The comments on GI.1 flexibility have been removed to avoid confusion on this point.  

3)    Lines 567-569. It is not clear what is meant by the epitope not being clearly defined. This goes back to the start of my comments. The escape mutations here are not, in the strictest sense, part of the epitope. The Ab/virus contacts are clearly defined by the cryo-EM structure and none of those residues were able to mutate to block antibody binding. This, again, shows the problem of defining an epitope by residues that affect antibody binding. I am not sure the best way to fix this other than to note at the start that epitope is being defined as residues that block antibody binding – but that these may include residues that do not directly contact the antibody but can cause conformational changes that do. In the absence of structural information on particular antibody/virus complexes, one cannot say for certain if those residues are part of the true definition of an epitope. 

Response:  Point taken.  As noted above, we have modified the text beginning at the new Line 230 to address the importance of structural information in the definition of an epitope. 

Reviewer 3 Report

I have now read the manuscript Viruses-493242 entiteled: “The Antigenic Topology of Norovirus as Defined by B and T Cell Epitope Mapping: Implications for Universal Vaccines and Therapeutics” by Jessica M. van Loben Sels and Kim Y. Green. The manuscript is a very well written and is an interesting overview and review over the published B and T cell epitopes on human norovirus proteins mapped, sofar. In my mind, the authors have done a splendid work and I have no further comments more than that I support the manuscript to be published in the Vaccines journal.

Author Response

 Reviewer 3

Comments and Suggestions for Authors

I have now read the manuscript Viruses-493242 entiteled: “The Antigenic Topology of Norovirus as Defined by B and T Cell Epitope Mapping: Implications for Universal Vaccines and Therapeutics” by Jessica M. van Loben Sels and Kim Y. Green. The manuscript is a very well written and is an interesting overview and review over the published B and T cell epitopes on human norovirus proteins mapped, sofar. In my mind, the authors have done a splendid work and I have no further comments more than that I support the manuscript to be published in the Vaccines journal.

Response:  We appreciate these comments.  Thank you.